# Towards system redesign: An exploratory analysis of neurodivergent traits in a childhood population referred for autism assessment

Jason Lang[1]*, Georgia Wylie[2], Caroline Haig[3], Christopher Gillberg[1], Helen Minnis[1]

**1** School of Health and Wellbeing, University of Glasgow, Glasgow, Scotland, **2** School of Medicine, University of Glasgow, Glasgow, Scotland, **3** School of Health and Wellbeing, Robertson Centre for Biostatistics, Glasgow, Scotland

* Jason.Lang@glasgow.ac.uk

**Data Availability Statement:** Data cannot be shared publicly because the data is owned by NHS Lanarkshire and is related to patient information. Data are available from NHS Lanarkshire by

## Abstract

### Background

Children's health services in many countries are moving from single condition diagnostic silo assessments to considering neurodevelopment in a more holistic sense. There has been increasing recognition of the importance of clinical overlap and co-occurrence of different neurotypes when assessing neurodivergent children. Using a cross-sectional service evaluation design, we investigated the overlap of neurodivergences in a cohort of children referred for autism assessment, focusing on motor, learning, and attention/activity level domains. We aimed to determine what proportion of children in a cohort referred for an autism assessment showed traits of additional neurodivergences, and what proportion were further investigated.

### Methods

We evaluated anonymised medical records of children aged between two and 17 years referred for autism assessment. We used validated questionnaires to assess for neurodivergent traits. A weighted scoring system was developed to determine traits in each neurodevelopmental domain and a score above the median was considered to indicate a neurodivergent trait. Evidence of further investigations were recorded. We then examined the relationships between autism traits and traits of additional neurodivergence.

### Results

114 participants were included for evaluation. 62.3% (n = 71) had completed questionnaires for analysis. Of these, 71.8% (n = 51) scored greater than the median for at least one additional neurotype, indicating the presence of other neurodivergent traits, and 88.7% (n = 64) attracted a diagnosis of autism. Only 26.3% of children with evidence of additional neurotypes were further investigated beyond their autism assessment.

contacting the Research and Development Manager (nhslresearch@lanarkshire.scot.nhs.uk) for researchers who meet the criteria for access to confidential data.

**Funding:** The author(s) received no specific funding for this work.

**Competing interests:** The authors have declared that no competing interests exist.

## Conclusions

Our results demonstrate the extensive overlap between additional neurodivergent traits in a population of children referred with suspected autism and show that only a small proportion were further investigated. The use of standardised questionnaires to uncover additional neurodivergences may have utility in improving the holistic nature of neurodevelopmental assessments.

## Introduction

Children's health services providing assessment or input for so-called neurodevelopmental conditions in Scotland, in line with the health services of many other nations [1], have been directed to move from single condition diagnostic silos to considering neurodevelopment in a more holistic sense [2, 3]. While potentially beneficial [4], such a system wide change can be challenging for services to implement, particularly in systems which were designed and staffed to examine separate single diagnoses [2].

Given recent movements in the neurodiversity community, such a change in the configuration of services should not be unexpected, and partly takes inspiration from the growth of acceptance which has developed around the neurodiversity paradigm [5]. However, in light of these new developments, there is a pressing need for services to better understand the needs of the neurodivergent population as a whole rather than, as previously, its parts.

The concept of neurodiversity was first discussed by Judith Singer in 1998 and was formalised in publication in 1999 [6]. In contrast to the predominant stream of earlier psycho-medical scientific discussion around neurodiverse variations, which labelled such conditions with pathological nomenclature and diagnosed them using a model of deficit, Singer proposed that neurodiversity was instead a natural state of the human population as a whole [6]. The concept shares commonality with the more widely understood idea of biodiversity [7], where a range of species contribute in their own ways to the health of an ecosystem. In the same way, Singer and other later proponents of the neurodiversity concept recognised that across a given population, there is a wide range of neuro-cognitive variability or styles [8], hereafter referred to as neurotypes or neurodivergences [9]. Each person's neurotype is unique, analogous to the concept of fingerprints.

Although there has, at least until recent times, been a limited academic discourse around the concept of neurodiversity, much has been written in "grey literature" and the concept has become embedded in autism self-advocacy forums and more recently in discussion forums around other neurodivergences [10]. Over the last few years, current thinkers have advanced the concept in academic terms, and this has given further legitimacy to the approach which was already largely accepted in neurodivergent communities. It is important that services reflect this change in culture and attitude in the approach which is taken towards neurodivergence.

Neurodevelopmental differences (traditionally pathologised using a deficit model) would be considered as forming part of this neurodiversity. Such neurotypes include so-called Autism Spectrum Disorder (referred to here as simply autism), Attention Deficit Hyperactivity Disorder (ADHD), Intellectual Disability (ID), and Developmental Coordination Disorder (DCD) among others [11]. If the premise of neurodiversity is correct, then the current systems approach, much prior research and DSM or ICD [12] diagnostic classifications may not be sufficient to recognise the complexity generated by different co-occurring neurotypes

during assessments. This is particularly likely to be the case in single diagnosis assessment services.

The work of Gillberg and colleagues on the concept of ESSENCE (Early Signs and Syndromes Eliciting Clinical Neurodevelopmental Examination) [13] may provide a bridge between these two approaches to neurodevelopment. Central to the ESSENCE concept is the idea of neurodiversity and the importance of clinical recognition of the overlap and co-occurrence of different neurotypes when developing formulations of neurodivergent children and young people. Much research demonstrates that if a person meets the diagnostic criteria for one neurotype, then it is likely that they may also meet criteria for, or at least have significant traits of, more than one [14]. However diagnostic manuals do not take neurotype *traits* into account given their categorical design. Even when significant traits which approach but do not cross the diagnostic threshold are present, applying categorical diagnostic techniques results in "no diagnosis", even if such neuro-minority traits may result in a child experiencing difficulties in their environment.

Neurodivergent may derive benefit from fully understanding their neuro-cognitive profile in greater detail than is provided by the application of simple diagnostic criteria [15, 16]. This may be particularly true where a person fully meets diagnostic criteria for one neurotype but presents with additional traits of other neurotypes which may play a role in the disposition of their unique neuro-cognitive style.

On the background of substantial change in both service structure and philosophical paradigms surrounding neurodevelopment, we wanted to better understand what likely neurotypes might be present in children who were referred to a service for autism diagnosis. All participants in this work were children referred for an autism assessment by a primary care service. However, given long waiting lists, these children had not been seen by the autism diagnostic service prior to it being incorporated into a larger more holistic neurodevelopmental service. Despite having been initially referred for an assessment solely for autism, their cases were instead dealt with by a new service established as per the Scottish Government's directive for approaching neurodevelopmental assessment holistically, as discussed above.

The aim of this cross-sectional service evaluation was to investigate the overlap of neurodivergences in a cohort of children referred for autism. We focussed on motor, learning, and attention/activity level domains, which are associated with DSM-5 diagnoses including so-called DCD, ID, and ADHD respectively. These neurotypes are common and have been found to co-occur with autism [17, 18].

Our questions were as follows:

1. What proportion of children in a cohort referred for an autism assessment had evidence of likely associated neurodivergences in addition to autism?

2. What proportion of children referred for an autism assessment received a diagnosis of autism, and what proportion were further investigated for another neurotype?

## Methods

Following Medical Research Council Guidelines, and discussion with NHS Research and Development colleagues, ethical approval was not required. The study was deemed a service evaluation of anonymised medical records of children who had been referred for clinical purposes. Caldicott Guardian approval for case note review was received.

Data was collected over a 6-month period in 2020 as part of clinical assessment. Participants included all children aged between two years and 17 years (n = 114) who were referred to the previous autism service but had yet to be seen at the time of its disbandment. All participants

had been referred with the primary clinical question of whether they met criteria for a DSM-5 diagnosis of autism, yet were seen clinically in the newly established neurodevelopmental service which aimed to assess a range of neurotypes in addition to autism.

As part of standard clinical assessment, parents or carers were asked to complete the 'Five to Fifteen' (FTF) [19] or 'Five to Fifteen–Toddler' (FTF-T) [20] assessment questionnaire. In addition to these questionnaires, each parent or carer completed a comprehensive developmental history and each child had at least one face-to-face clinical assessment session with an experienced practitioner. This clinical assessment process was in line with national guidelines [21].

The FTF and FTF-T questionnaires are validated questionnaires [22–25] which ask parents about symptoms of all neurodivergences included under the ESSENCE umbrella, as well as highlighting a child's specific strengths and weaknesses within different domains [26]. While the FTF and FTF-T questionnaires are similar, they contain different questions to represent the stages of development within each age group (two to five years and five to 17 years). Both questionnaires are composed of ten sections which focus on different areas of development.

The FTF and FTF-T questionnaires were not originally created with a scoring system and are based on symptoms and functional criteria rather than diagnostic criteria. These questionnaires had been introduced into routine clinical practice within the new neurodevelopmental service. In order to conduct this evaluation, a weighted scoring system was developed, with permission of the questionnaire authors (see below, in Results), to indicate the degree of impact recorded in each neurodevelopmental domain.

To examine neurodivergent traits in addition to autism, we focussed on three additional domains encompassing common neurotypes: attention/ activity levels, motor development, and learning, since the DSM-5 diagnoses most closely corresponding to these neurotypes, ADHD, DCD and ID, have previously been found to be associated with autism [17].

Cases were split into three age groups: two-five years; five to eight years; and eight to 17 years. The demographic data for each case was entered into one of three data collection tools based on age group. Questionnaire scores were also entered, allowing scores for each developmental domain to be calculated.

Relevant questionnaire items were combined to produce domain scores. for the *ADHD-like domains*, items from the following questionnaire sections, 'Attention and concentration, Planning and organisation', 'Overactivity and impulsivity', and 'Memory', were combined. The "Gross and fine motor skills" section was used to produce the *motor domain* scores. The 'Learning/Acquisition of academic skills' section was used to produce the s *learning domain* scores. Learning/Academic domains could only be analysed in the two to five and eight to 17 age groups as the questionnaires did not include a learning section for the five-eight age group.

For each domain, a weighted score was calculated by taking the child's domain score and multiplying this by the functional impairment score for that domain. This was then standardised by dividing by the maximum domain score. The median and mean weighted scores of the whole cohort who completed the questionnaires, and for children with completed questionnaires and diagnosed autism, were recorded for each of the three additional domains of interest.

A questionnaire score above the median for the whole cohort was taken to indicate the likely presence of traits of an additional neurotype. As noted above, while the presence of such traits does not necessarily indicate an additional diagnosis, application of the scoring method described ensured that both the presence and impact of traits within each additional domain was accounted for.

Final clinical diagnosis of autism at completion of assessment was documented for each case, as well as whether there were any steps taken to further investigate any potentially indicated neurotypes.

Minitab was used for statistical analysis. A $\chi^2$ test of association was used to analyse the association between the proportion of children with scores above the median in the various neurotype groups and age group; the number of additional likely neurotypes; final autism diagnosis and age group; the proportion of males and the number of additional neurotypes detected. Prior to statistical analysis it was agreed that the median for the whole cohort would be used to indicate the likely presence of each neurotype because the distribution of the motor scores were skewed (Mean = 0.7988 Median = 0.5).

A two-sample proportion test was used to analyse the difference between the proportion of all children with evidence of additional neurotypes and those with autism and evidence of additional neurotypes. A one-way ANOVA was used to analyse the relationship between additional neurotypes and age at assessment. Case note data was pseudonymised prior to analysis.

## Results

Whole cohort medians from extracted questionnaire data were as follows:

- ADHD-like domain = 4.8 (out of 9).

- Motor domain = 0.5 (out of 3).

- Learning domain = 1.345 (out of 3).

Children with a score greater than the median score were deemed to have traits in this functional area. The median score for the whole cohort and the median for the cohort with autism were not statistically different for any of the domains.

The demographics of the participants are shown in Table 1 and the participant flow diagram can be seen in Fig 1. A total of 114 children's records were included in the analysis. There was a higher proportion of males referred, and this was particularly apparent in the youngest age group (85.3% male in the two-to-five-year age group).

Of all records examined, 78.9% met the diagnostic criteria for autism and were given a diagnosis. Younger children were significantly more likely to be diagnosed with autism than their older peers (p = 0.028).

In total 62.3% (n = 71) of participants had completed FTF or FTF-T questionnaires available for analysis in their case notes. There was no statistical difference between the with-questionnaire and without-questionnaire groups in terms of recorded demographics. Of the

**Table 1. Participant demographics.**

|  | Ages 2–5 | Ages 5–8 | Ages 8–17 | Overall |
|---|---|---|---|---|
| **Demographics** | | | | |
| Number of children (%) | 34 (29.8%) | 22 (19.3%) | 58 (50.9%) | 114 (100%) |
| Mean age at time of assessment in months (SD) | 54.6 (6.01) | 77.7 (10.6) | 134.6 (27.9) | 97.8 (41.7) |
| Male (%) | 29 (85.3%) | 15 (68.2%) | 45 (77.6%) | 89 (78.1%) |
| Female (%) | 5 (14.7%) | 7 (31.8%) | 13 (22.4%) | 25 (21.9%) |
| Mean waiting time from referral to assessment in months (SD) | 17.1 (4.4) | 18.2 (4.2) | 18.2 (6.1) | 17.8 (5.3) |
| No. of completed questionnaires | 23 (67.6%) | 17 (77.3%) | 31 (53.4%) | 71 (62.3%) |
| **Diagnosis and investigation** | | | | |
| No. diagnosed with autism (%) | 30 (88.2%) | 20 (90.9%) | 40 (69%) | 90 (78.9%) |
| No. investigated for another condition (%) | 10 (29.4%) | 10 (45.5%) | 10 (17.2%) | 30 (26.3%) |

Table 1 shows the demographics of the participants (%) by age group and total, including sex, waiting tome for assessment, number of completed questionnaires, autism diagnosis and number (%) of those investigated for other neurotypes in addition to autism.

**Fig 1. Participant flow diagram.** Fig 1 shows the numbers (%) of referred children, with completed data, numbers (%) eventually diagnosed with autism and numbers (%) with evidence of additional neurodivergence.

with-questionnaires group, 88.7% (n = 64) of children with a completed questionnaire attracted a clinical diagnosis of autism and 71.8% (n = 51) had scores greater than the median in at least one additional neurotype group. The percentage of children with one or more indicated neurotypes was highest in the two-to-five-year age group although this may have been an effect of sex.

Of the children who attracted a diagnosis of autism and had a completed questionnaire (n = 63), 76.2% (n = 48) had questionnaire evidence of one or more additional neurotypes (see Fig 2): 55.6% (n = 35) had evidence of a possible ADHD-like neurotype; 52.4% (n = 33) had indications or a neurotype on the motor domain; and 36.5% (n = 23) had indications of a possible learning difficulty or disability. As can be seen from Fig 2, these additional neurotypes overlapped with each other as well as with autism.

Motor domain-associated neurotypes were more common in younger children while ADHD-like domain-associated neurotypes were more common in the older population. Children with more than one possible neurotype in addition to autism were significantly more likely (p = 0.017) to be assessed at a younger age.

Full results concerning likely additional neurotypes discovered in children with completed questionnaires can be seen in Table 2.

After completing autism assessment, 26.3% of children had case-note evidence of formal investigation of other possible underlying neurotypes in addition to their autism diagnosis.

## Discussion

Our results demonstrate that a significant proportion (76.2%) of children referred for an autism assessment in this cohort demonstrated evidence of at least one other underlying

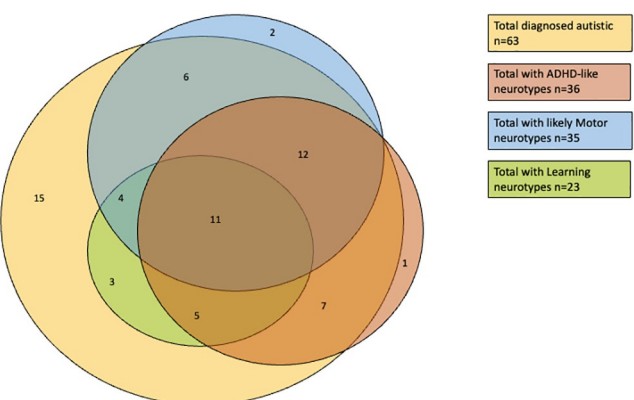

**Fig 2. Overlap of identified neurotypes in a cohort of children referred for Autism assessment (n = 66).** The numbers in segments relate to the children identified with these specific neurotypes. n = 15 children were identified as only autistic. n = 2 were identified as having only a motor neurotype. n = 1 was identified as having only an ADHD like neurotype. All children identified as having a learning neurotype also had other neurotypes present. The numbers in each segment refer to children with more than one overlapping neurotype and reflect the corresponding overlaps.

neurodivergence in addition to an autism diagnosis, however relatively low numbers of these children (26.3%) were formally investigated for an additional underlying diagnosis. In addition, we report that nearly 80% of the children referred for an autism assessment, in this sample, met the diagnostic criteria as defined in DSM-5 after face to face clinical assessment.

Our findings show extensive overlap between autism and additional neurodivergences in ADHD and/or motor and/or learning domains. Our findings support the ESSENCE concept [13] and are in line with a growing research base illustrating common genetic, anatomic and environmental factors which may be responsible for phenotypic overlap between neurodivergences [27, 28].

Our results indicate that up to 55.6% of children referred for autism assessment may also meet the diagnostic threshold for ADHD, and certainly have at least some significant ADHD traits. The proportion is markedly higher than ADHD prevalence in the general population,

**Table 2. Additional neurodivergent traits discovered by age and total for whole cohort.**

|  | Ages 2–5 | Ages 5–8 | Ages 8–17 | Total |
|---|---|---|---|---|
| **Additional neurodivergent traits in children with completed questionnaires** | | | | |
| No. with completed questionnaires | 23 | 17 | 31 | 71 |
| No. with **ADHD-like** neurotype evidence (%) | 10 (43.5%) | 11 (64.7%) | 15 (48.4%) | 36 (50.7%) |
| No. with **motor domain** neurotype evidence (%) | 17 (73.9%) | 9 (52.9%) | 9 (29.0%) | 35 (49.3%) |
| No. with **learning domain** neurotype evidence (%) | 11 (47.8%) | n/a | 13 (41.9%) | 24 (33.8%) |
| Total with additional neurotypes in **one or more domains** | 19 (82.6%) | 13 (76.5%) | 19 (61.3%) | **51 (71.8%)** |
| **Additional neurodivergent traits in children with completed questionnaires AND diagnosed autism** | | | | |
| No. with completed questionnaires and autism diagnosis | 21 | 15 | 27 | 63 |
| No. with **ADHD-like** neurotype evidence (%) | 10 (47.6%) | 11 (73.2%) | 14 (51.9%) | 35 (55.6%) |
| No. with **motor domain** neurotype evidence (%) | 16 (76.2%) | 9 (60.0%) | 8 (29.6%) | 33 (52.4%) |
| No. with **learning domain** neurotype evidence (%) | 11 (52.4%) | n/a | 12 (44.4%) | 23 (36.5%) |
| Total with additional neurotypes in **one or more domains** | 17 (81.0%) | 13 (86.7%) | 18 (66.7%) | **48 (76.2%)** |

Table 2 shows the number (%) of neurodivergent traits detected in all participants with completed questionnaires and in those with completed questionnaires and a diagnosis of autism, by age.

which is estimated to be 5–7% in children [29]. Since missed or late diagnosis of ADHD is associated with an increased risk of negative psychosocial outcomes such as substance misuse and criminality across the lifespan [30], these data provide further evidence for the utility of holistic neurodevelopmental assessments.

Autism and ADHD commonly co-occur in children [31–35], with genetic components influencing 53 separate neurodevelopmental symptoms across conditions in one study [27]. Family and twin studies support the idea that these neurotypes share a genetic underpinning. However, the specific genetic, structural or environmental mechanisms contributing to the coexistence and overlap of these neurotypes remains to be fully elicited [16, 36]. There is also evidence that ADHD symptoms may manifest across other diagnostic categories such as Intellectual Disability and autism, further emphasising the need to avoid rigid adherence to current diagnostic boundaries but, instead, to be led by traits and functional impairment.

Our results showed that 36.5% of our cohort exhibited some evidence of learning differences. Aside from a potential overlap with underlying Intellectual Disability (which if the case would be grossly overrepresented in this group compared to community prevalence of 3.2%) [37], specific learning differences may also co-occur with other neurodivergences. For example, dyslexia has been shown to closely associate with several other neurodivergences including ADHD and developmental motor differences [38].

Similarly, possible neurodivergences in motor domains (52.4%) are over-represented here compared to community prevalence rates for the corresponding DSM-5 diagnoses (DCD 1.8%) [17]. Motor differences can occur as part of the autistic phenotype [39], as part of other neurodivergences [40], or as a discreet neurodivergence [41]. Motor differences can be associated with significant functional difficulties, but are rarely considered in child mental health clinics [42, 43]. Our evidence of a possible higher frequency of underlying motor neurotypes in our population of autistic children, compared with the general population, further highlights the overlapping nature of neurodivergence and the importance of holistic assessment strategies. Motor neurodivergence may otherwise go unrecognised [41].

Although we only focussed on the three most prevalent additional neurodivergences, our results are in close agreement with previous findings that 70% of children with autism will have one or more co-occurring neurodivergence [17]. Even if only a small proportion of the children with possible additional neurodivergences in our cohort would go on to meet diagnostic threshold for the associated DSM-5 conditions, detecting traits of such neurodivergences is important in developing personal, family, and systemic insight about the child.

From our review of the extant literature, we were not able to identify any studies which contradicted our findings, however there is reported variability in the rate of detected neurodivergences reported in other work. Our rates of possible underlying neurodivergence are higher than those of some other studies which have measured co-occurrence of diagnoses as defined in DSM-5. Zauche and colleagues, for example, found approximately 21% of their sample of autistic children also exhibited symptoms of ADHD. Furthermore, cognitive impairments were present in 18% of autistic children [44]. However, their study was based on a retrospective review of case notes and did not employ holistic questionnaires to gather evidence. The holistic approach taken here is a relative strength of our study.

Our results show a positive association between the number of neurodivergences detected and an earlier age of referral. This may add weight to our contention that the presence of additional traits of neurodivergence may be important in better understanding the impact of complexity and possible associated disability in this population. When co-occurrence is present, it can lead to increased disability and may necessitate multiple intervention strategies [45]. Co-occurring autism and ADHD have also been associated with an increased risk of mood and anxiety disorders [46].

Of the children who had a completed questionnaire, 88.7% were diagnosed with autism. This was higher than the percentage who were diagnosed from the overall referral cohort, at 78.9%. The difference between these percentages may suggest parents who are more concerned about their child may be more likely to complete the questionnaire. Conversely, it might suggest that the questionnaire was more likely to help clinicians identify children with autism. Dewey, in her review of co-occurring neurodivergence, concluded that such co-occurrence may simply reflect the inability of current categorical diagnostic systems to fully recognise the profiles of specific neurodivergences [47]. Using a less categorical approach to understand neurodivergent children and young people is also suggested by Rivard and colleagues following their cluster analysis into overlapping presentations [48]. The use of holistic tools which concentrate on traits rather than diagnostic criteria may therefore be clinically useful.

In comparison to the potential co-occurrence which we have demonstrated, the percentage of children who were formally investigated for possible additional neurodivergence was only 26.3%. Once diagnosed with autism, children may reach the end of a 'pathway' and the presence of other neurotypes may not be acknowledged or investigated. Even if children are investigated further after being diagnosed with one neurodivergence, this can often entail a lengthy and stressful additional process [4] where services are not integrated. Our research further adds to the emerging evidence base suggesting that integrated systems are likely to be of more benefit to children, young people, and their families [49–52].

Currently, the majority of health service structures, at least within the UK if not internationally, do not accommodate the neurodivergent cases that these children demonstrate [4]. This may in part be due to a historical over-reliance on diagnostic criteria which fail to capture complexity and overlap [53]. To better support neurodivergent children, health services should anticipate children having multiple neurodivergences and respond appropriately [50, 54]. Some models in the UK have shown that integrated services can be provided, with promising results including shorter waiting times and reduced stress for children and their families [4, 55]. More research into the effectiveness of this integrated approach is needed.

Our results suggest the potential clinical utility of routinely using validated questionnaires to holistically screen children referred to services with a possible neurodivergence. Potential cases would most likely have gone undetected if only autism specific assessment measures had been applied in this population.

In common with other literature in this area, we found that the percentage of females referred with suspected autism was notably lower than males. Globally the majority of people, at least with diagnosed neurodivergences, are male [13]. While males may be more neurobiologically likely to develop neurodivergence [56] it may also be that these neurodivergences remain under-detected in females [17, 56]. Of course, it may be that current neurodevelopmental assessment tools are less sensitive to female presenting characteristics, with some evidence suggesting that females may have a different autistic profile compared to males [57]. This, coupled with possible clinician and parental biases [58], may result in diagnoses in female patients being missed more often than in males.

Our results suggest that neurodivergent females were less likely than males to be identified before the age of five. A smaller percentage of females were referred in the 2–5 age group than in the older age groups. Other studies have shown that it is less common for females to be identified before school-age in comparison to males [13, 58]. Previous studies have demonstrated that autism and ADHD are more likely to co-occur in males than females [45]. Other research has found that females are more likely to have co-occurring features of autism and ID than males [59]. It may be that some conditions are more likely to co-occur in males than in females, and vice versa, and the presentation of these overlaps may influence the age at which

these neurodivergences come to light. Further research should examine this relationship in larger cohorts.

Children with more complexity were significantly more likely to be assessed for autism at a younger age. This agrees with a previous study carried out using the FTF [26]. A child whose problems are neurodevelopmentally complex may be more noticeable to parents and clinicians, which may explain why they are assessed for neurodivergence at an earlier age than a child with less complexity. This is potentially another reason why females, with seemingly less complexity, are identified at older ages than males.

This study has some important limitations which may have affected our results. The relatively small cohort size may have obscured important findings about subgroups. Cohort size was further limited by questionnaire completion rates of 62.3%. The clinical audit design meant that we were unable to influence the rate of completion of questionnaires. It may be that children with completed questionnaires were more or less likely to have functional impairments than those without, and therefore it is possible that the percentage of children with functional impairments is different than the percentage we found. In addition, our questionnaires did not have a section on learning domains for the five-to-eight age group, so functional impairment in learning domains could not be obtained for this group. This paper only focussed on additional neurotypes associated with learning, motor and ADHD-like domains and did not account for the overlap of other neurodivergences with autism.

The presence of likely neurodivergences was implied by using the cohort median as a cut-off since data from the general population was not available with which to create population norms and, in any case, such norms may not have been applicable to this clinical group. The cut-off score was based on a cohort of children all of whom had likely neurodivergences, so it is likely that the percentage of children with neurodivergent traits with respect to the general population is actually higher than the result shown here. Additionally, this sample was not ethnically diverse and so we were not able to observe any associations with ethnicity. Finally, there was no reliable way to obtain information on possible underlying carer stress or carer neurodivergence which may have altered results. This should be considered in future research in this area.

## Conclusion

In our population, three out of four children with autism (76%) had evidence of one or more additional neurodivergence, yet only 26% were referred on for additional assessments. At present, most health services do not accommodate these co-occurring neurodivergences and either treat them as separate entities or overlook them completely. This level of co-occurrence and overlap may have significant implications for how future services are designed for this group of children. Validated questionnaires may help clinicians to identify co-occurring neurodivergences at the first assessment, allowing for earlier support and the development of whole system insight to a child's neurotype.

Future research should look in more depth, and with a greater range of validated assessment methods, for further associations across neurotypes in this population group. As this preliminary work only concentrated on three possible neurodivergences, adding further measures to future prospective studies, including additional neurodivergences and mental health outcomes, would also be welcomed. While this work is based in children's services, we would also recommend that similar studies be carried out in adult populations, where current approaches remain, to a large extent, siloed in approach.

## Supporting information

**S1 Checklist. STROBE statement—Checklist of items that should be included in reports of observational studies.**
(DOCX)

## Author Contributions

**Conceptualization:** Jason Lang.

**Data curation:** Jason Lang, Georgia Wylie, Caroline Haig.

**Formal analysis:** Georgia Wylie.

**Investigation:** Georgia Wylie.

**Methodology:** Jason Lang.

**Supervision:** Jason Lang, Christopher Gillberg, Helen Minnis.

**Validation:** Caroline Haig.

**Writing – original draft:** Jason Lang.

**Writing – review & editing:** Jason Lang, Georgia Wylie, Caroline Haig, Christopher Gillberg, Helen Minnis.

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
