## [Decision Letter · Decision Letter 0]

15 Oct 2023

PONE-D-23-18648Towards system redesign: an exploratory analysis of neurodivergent traits in a childhood population referred for autism assessment.PLOS ONE

Dear Dr. Lang, 

Thank you for submitting your manuscript to PLOS ONE. After careful consideration, we feel that it has merit but does not fully meet PLOS ONE’s publication criteria as it currently stands. Therefore, we invite you to submit a revised version of the manuscript that addresses the points raised during the review process.

The Academic Editor concurs with the observations made by the reviewers regarding Figure 2, which presents challenges in comprehension. It is imperative to enhance the clarity of the figure legends, ensuring that they are explicitly labeled in accordance with their respective sequencing. Moreover, in the ensuing discussion, there arises the necessity to incorporate a more substantial body of evidence derived from the extant scholarly literature. Additionally, it is advisable to scrutinize whether any contrasting studies exist in the relevant literature that merit consideration and inclusion within the discourse 

We look forward to receiving your revised manuscript.

Kind regards,

Asrat Genet Amnie, MD, EdD, MPH, MBA

Academic Editor

PLOS ONE

3. Please remove your figures from within your manuscript file, leaving only the individual TIFF/EPS image files, uploaded separately. These will be automatically included in the reviewers’ PDF.

Additional Editor Comments:

This Academic Editor concurs with the observations made by the reviewers regarding Figure 2, which presents challenges in comprehension. It is imperative to enhance the clarity of the figure legends, ensuring that they are explicitly labeled in accordance with their respective sequencing. Moreover, in the discussion section, there arises the necessity to incorporate a more substantial body of evidence derived from the extant scholarly literature. Additionally, it is advisable to scrutinize whether any contrasting studies exist in the relevant literature that merit consideration and inclusion within the discourse.

Reviewers' comments:

Reviewer's Responses to Questions

**Comments to the Author**

1. Is the manuscript technically sound, and do the data support the conclusions?

Reviewer #1: Yes

Reviewer #2: Yes

2. Has the statistical analysis been performed appropriately and rigorously? 

Reviewer #1: Yes

Reviewer #2: Yes

3. Have the authors made all data underlying the findings in their manuscript fully available?

Reviewer #1: Yes

Reviewer #2: Yes

4. Is the manuscript presented in an intelligible fashion and written in standard English?

Reviewer #1: Yes

Reviewer #2: Yes

5. Review Comments to the Author

Reviewer #1: This research is of importance as it builds evidence for the benefits of a neurodevelopmental pathway rather than siloed services focussing on one condition at a time. The authors have acknowledged that this is a small sample but the findings are unequivocal and make a powerful case of looking for commonly coexisting neurodevelopmental conditions

The methodology is clearly explained and the discussion is relevant and clear.

However

I found Figure 2 difficult to understand (line 270)

The statement about the selection of median needs a little more explanation. In what ways were the mean scores skewed (line 213-214)

In the final paragraph Line 418 onwards) the authors may consider recommending a similar study in the adult population as they too are not well served by siloed services, a trend that is increasingly prevalent.

Reviewer #2: Thankyou for the invitation to review the manuscript.

Overall this is an interesting study. The article is presented in an intelligible fashion and is written in standard English.

It can be accepted with a few changes

1. Legend of figures need to be more clear and labelled as per sequence.

2. 2nd figure in the sequence is not clear

3. Discussion should include more evidence from literature and if there are any contrasting studies?

4. Page 301: 'only' should be removed

5. Page 338: Replace caseness with 'cases'

6. Use 'DSM-5' instead of 'DSM5' throughout the article

6. PLOS authors have the option to publish the peer review history of their article (what does this mean?). If published, this will include your full peer review and any attached files.

Reviewer #1: No

Reviewer #2: **Yes: **Prof Dr Shemaila Saleem

---

## [Author Response · Author response to Decision Letter 0]

26 Nov 2023

We thank the editor and reviewers for their very helpful comments. We have adopted the comments in full and have detailed this in the accompanying Response to Reviewers letter as uploaded.

---

## [Editor Report · Decision Letter 1]

6 Dec 2023

Towards system redesign: An exploratory analysis of neurodivergent traits in a childhood population referred for autism assessment.

PONE-D-23-18648R1

Dear Author, 

We’re pleased to inform you that your manuscript has been judged scientifically suitable for publication and will be formally accepted for publication once it meets all outstanding technical requirements.

Kind regards,

Asrat Genet Amnie, MD, EdD, MPH, MBA

Academic Editor

PLOS ONE

Additional Editor Comments (optional):

The authors appear to have fully addressed the concerns raised and the comments made by the reviewers.
---

## [Editor Report · Acceptance letter]

12 Dec 2023

PONE-D-23-18648R1 

PLOS ONE

Dear Dr. Lang, 

I'm pleased to inform you that your manuscript has been deemed suitable for publication in PLOS ONE. Congratulations! Your manuscript is now being handed over to our production team.

Kind regards, 

on behalf of

Dr. Asrat Genet Amnie 

Academic Editor

PLOS ONE